# Use of Bethanechol, 50 mg/BID, for a Failed Decatheterization Test: A Position Statement

**James Walter [1,2,*] and John Wheeler [1,3]**

1   Emeritus, Department of Urology, Loyola Medical Center, Maywood, IL 60153, USA; JWheele77@gmail.com
2   Emeritus, Research Service, Hines VA Hospital, Hines, IL 60141, USA
3   Emeritus, Department of Surgery, Hines VA Hospital, Hines, IL 60141, USA
*   Correspondence: JamesWalter889@gmail.com

**Abstract:** If urinary catheters are used during surgeries, epidurals and parturition, patients and their physicians want a return to volitional voiding as soon as possible. Micturition is regained by passing a decatheterization test. Fortunately, only a small percentage of patients fail three or more of these tests and have to use catheters longer-term. Surprisingly, there are no approved drugs that are currently approved to assist with these tests; however, bethanechol, 50 mg/BID, should be considered further for this application. The drug is a bladder wall receptor stimulant and, at this moderate dosage, it is reasonable to expect it to help with decatheterization tests. This position statement includes several designs for future bethanechol use and research. In addition, an International Neuro-Urological Research Group is introduced that is promoting this drug.

**Keywords:** lower urinary tract; urinary incontinence; urinary retention; catheter; urinary tract infections





## 1. Introduction

Foley catheters are often used during surgeries, parturition and epidurals; however, catheters are associated with discomfort and the risk of catheter-associated urinary tract infections (CAUTI) [1]. To stop using catheters and return to volitional voiding, the patient must pass a decatherization test. This test is conducted when the risk of postoperative urinary retention (POUR) is low. After a transurethral resection of the prostate, the test is usually conducted when hematuria is no longer present [2,3]. After radical prostatectomy, there is a typical delay of 5 to 12 days to obtain a secure bladder neck anastomosis. However, with laparoscopic methods, safe decatheterization tests are reported on postoperative Day 2 to 4 [4]. Moreover, 12 to 45% of women will experience postpartum urinary retention needing catheterization, and the first decatheterization test is usually conducted on the same or the next day [5]. A similar decatheterization time exists after epidural anesthesia, and other surgeries including anorectal, orthopedic and inguinal [6].

For the decatheterization test, the bladder will first be drained and then the catheter will be removed. The patient will remain in the clinic for 4 to 5 h while the bladder fills naturally at 1 mL/min, reaching a volume of 240 to 300 mL. Ultrasound can be used to insure that overdistention of the bladder does not occur, because bladder contractility decreases after a filling volume of 300 mL, and 450 mL is considered overdistention [6]. The time spent in the clinic can be reduced by filling the bladder with sterile isotonic saline prior to catheter removal. During the clinic time, the patient is asked to try to void, and this can be facilitated by the sound of running water. Volitional voiding is demonstrated when 100 mL or more is voided. If the patient is unable to urinate, demonstrating POUR, recatheterization is needed and additional tests will be conducted at a later date [6]. The time between tests allows for more recovery of bladder contractility and urgency, both of which are needed for volitional voiding. Most patients pass their first test and 95% or more will pass within three tests. The remaining 5% or less, however, will continue to need to use catheters. The primary risk factor for failing multiple tests is overdistention or the bladder

prior to surgery, denervation or urethral injury during childbirth. Secondary risk factors include advanced age, obesity and smoking [3,4,6].

Failed decatheterization tests are associated with POUR and the condition of an underactive bladder (UAB), also named bladder acontractility, and/or a lack of urgency [4,6]. Some medication groups such as alpha blocking and anticholinesterases are used off-label to help patients pass decatheterization tests and treat UAB.

Bethanechol (urecholine) should also be considered to help patients pass decatheterization tests, and we were surprised to find no reports in the literature about this use. The drug induces detrusor (bladder wall) contraction through activation of the acetylcholine muscarinic receptor subtype that is located in both the lower urinary tract and colon [7–10]. The primary reason why bethanechol is not being used is that it has not demonstrated efficacy in patients who are voiding but have an UAB condition with high residual volumes. Urecholine in dose–response tests for UAB has failed to increase bladder pressures and flow, or reduce residual volume [7–10]. According to the International Continence Society guidelines, bethanechol should not be prescribed in cases of non-obstructive urinary retention and UAB. With these limitations in mind, here are two reasons why bethanechol should be considered for failed decatheterization tests and POUR. First, a failed test is caused not only by UAB but also by a lack of voiding urgency. There is strong evidence that bethanechol increase urgency, as demonstrated in cystometric studies [8,9]. Second, weak evidence comes from a conversation with two urologists practicing in Chicago, United States. They stated that oral bethanechol at 50 mg BID had helped their patients pass decatheterization tests, and that lower doses were ineffective and higher doses increased the risks of side effects (unpublished data). They stopped the drug after volitional voiding occurred; however, if POUR returned, the drug was restarted with the return of volitional voiding. They gave occasional drug holidays to ensure that the drug was needed. No patient had stopped the drug because of side effects, such as diarrhea, respiratory secretions, or bradycardia. The current authors of this paper, however, believe that caution should be exercised in the use of bethanechol particularly in the short postoperative period. The use of this drug will depend on future research as outlined below.

## 2. Future Research

Retrospective case or case series on the use of bethanechol (50 mg/BID) related to decatheterization tests are needed. For this design, standard clinical care should be followed and a case report considered after the course of therapy. There are several options about when the drug could be started. An early application would be toward the end of the first decatheterization test; if it looked like it was going to be a failed test, it might convert the test to a positive response. The drug could also be started after the first failed test, or the clinician might decide to wait and start the drug after a second or third failed decatheterization test, because most patients go on to pass these tests without drug assistance. When to stop the drug would also depend on clinical decisions. The expectation would be to stop the drug as soon as the patient demonstrates volitional voiding; however, because of the risk of reverting to POUR, this could be delayed. Decisions about bethanechol timing should take the risk of side effects of the drug and CAUTI into consideration, and this has become more serious because of increased resistance to antibiotics [11]. In addition, during the process of multiple decatheterization tests, the type of catheter is usually changed from Foley to intermittent (IC) to reduce the risk of CAUTI [11].

Prospective case series design usually require approval and close coordination with an institutional review board (IRB) including informed consent. In the United States, bethanechol is a prescription drug with indications for both UAB and urgency, meaning that it would not be an off-label use; thus, a quality-of-care committee could also approve a bethanechol study [4]. For this design, the timing of bethanechol will usually be the same for all patients because uniform treatment increases the likelihood of obtaining conclusions with higher validity.

Clinical trial research is known as a high-quality design, and this type of design is usually needed to change clinical practice. Close coordination with an IRB is required, including informed consent. Patients would usually be randomized to a control group receiving standard care or an experimental group receiving standard care plus bethanechol. The time to start and stop the drug would be specified. A concern about clinical trial research is its high cost. We contacted bethanechol's manufacturers and distributors about research for this drug, and they declined to be involved because of the drug's generic status and the high cost of the research.

**Author Contributions:** Concept, writing, review and editing—J.W. (James Walter) and J.W. (John Wheeler). All authors have read and agreed to the published version of the manuscript.

**Funding:** This research received no external funding.

**Institutional Review Board Statement:** Not applicable.

**Informed Consent Statement:** Not applicable.

**Data Availability Statement:** Not applicable.

**Acknowledgments:** We thank Achim Herms for his guidance in this submission. The current authors are members of an international neuro-urology research group, which was formed with the goal of fostering research in this area. We hope this publication will foster further bethanechol use and research; contact James Walter, at JamesWalter889@gmail.com.

**Conflicts of Interest:** The authors declare no conflict of interest.

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
