# Peer review of "Use of Bethanechol, 50 mg/BID, for a Failed Decatheterization Test: A Position Statement"

_2673-4397, doi:10.3390/uro2020016_

Round 1

Reviewer 1 Report

At the outset, I would like to congratulate the authors for submitting this opinion article. The work has merit and will be of interest to our readers.

I have a few comments that will improve the quality of this manuscript.

Please compare Urecholine with other modalities used for failed decatheterization. This can be highlighted in one paragraph

Also, please correct the grammatical errors. There are major grammatical mistakes that need a writing assistant.

In addition, please remove the section numbering. It is not required. Section 2 does not depict the study designs. In fact, it gives pointers for future research. 

Section 3 can be added in the acknowledgments. 

Author Response

Reviewer 1, We thank the reviewer for their very helpful comments. We have extensively edited and revised the submission as requested, including an editor’s assistance.

Concern 1: Please compare Urecholine with other modalities used for failed decatheterization. This can be highlighted in one paragraph. Thank you, we added other modalities in sentences: Some medications groups such as alpha blocking and anticholinesterases are used off-label to help patients pass decatheterization tests and treat UAB. According to the International Continence Society guidelines, Bethanechol should not be prescribed in case of non-obstructive urinary retention, and the drug needs to be used off-label.

Concern 2: Also, please correct the grammatical errors. There are major grammatical mistakes that need a writing assistant. We hired a writing assistant, and we have corrected grammar.

Concern 3: In addition, please remove the section numbering. It is not required. Section 2 does not depict the study designs. In fact, it gives pointers for future research. We agree and have made the changes. We changed the second section title to Future Research.

Concern 4: Section 3 can be added in the acknowledgments. We agree and have made the changes

Reviewer 2 Report

Thank you for considering me as a reviewer for this publication. I have provided my comments as follows.

General Comments:

The authors described the use of bethanechol in patients with post-operative urinary retention, which contributes to existing knowledge, and is especially important because very little has been written about it in the scientific literature.

The etiology of post-operative urinary retention should always be considered. Whether it is a transient effect of anesthetics or denervation of the bladder during pelvic surgery or damage to the urethra during eg vaginal birth.

These drugs have been used for a long time in clinical practice in patients with non-obstructive urinary retention. Their effects on bladder emptying have been described in the literature. Unfortunately, their action is usually not 100% effective and after some time the effect weakens. Dangerous side effects such as bradycardia are also possible.

The indications for prescribing these drugs in some states are not intended to empty the bladder but in cases of constipation. Also, according to the EAU and ICS guidelines, due to the questionable effect, these drugs should not be prescribed in case of non-obstructive urinary retention. Therefore, these drugs are prescribed off-label. As a clinician, I believe that caution should be exercised in the use of these drugs even in the short post-operative period, and potential adverse events should be considered. There is a lack of knowledge among urologists about the possibility of using these drugs, so any article on this topic is welcome. Especially if a larger randomized study is planned.

This article can be accepted for publication but some spelling mistakes need to be corrected (see the Specific comments).

This article will be useful for surgeons, urologists, and gynecologists.

Specific Comments:

Abstract

Bethanechol is sold under many brand names, amongst them is Urecholine (Merck Frosst). I presume thet Urecholine is a protected name, and I think that it might be better to use a generic name (Betanechol) in the text.

Study designs

Line 94. Reporing = Reporting

Author Response

Reviewer 2, We appreciate the review and the helpful comments.

General Comments: The authors described the use of bethanechol in patients with post-operative urinary retention, which contributes to existing knowledge, and is especially important because very little has been written about it in the scientific literature. The etiology of post-operative urinary retention should always be considered. Whether it is a transient effect of anesthetics or denervation of the bladder during pelvic surgery or damage to the urethra during eg vaginal birth.

Response: we appreciate and agree with these comments. We changed neural injury to denervation

Comment 2. These drugs have been used for a long time in clinical practice in patients with non-obstructive urinary retention. Their effects on bladder emptying have been described in the literature. Unfortunately, their action is usually not 100% effective and after some time the effect weakens. Dangerous side effects such as bradycardia are also possible. The indications for prescribing these drugs in some states are not intended to empty the bladder but in cases of constipation. Also, according to the EAU and ICS guidelines, due to the questionable effect, these drugs should not be prescribed in case of non-obstructive urinary retention. Therefore, these drugs are prescribed off-label.

As a clinician, I believe that caution should be exercised in the use of these drugs even in the short post-operative period, and potential adverse events should be considered.

There is a lack of knowledge among urologists about the possibility of using these drugs, so any article on this topic is welcome. Especially if a larger randomized study is planned.

Response: we appreciate these comments. We added bradycardia as a risk.

Specific Comments: Abstract - Bethanechol is sold under many brand names, amongst them is Urecholine (Merck Frosst). I presume thet Urecholine is a protected name, and I think that it might be better to use a generic name (Betanechol) in the text. Study designs Line 94. Reporing = Reporting

Response: We have made these changes; Bethanechol is now used.

Reviewer 3 Report

I agree with your suggestion that Urecholine can be consider as a useful agent for improving decatherization.

Author Response

Reviewer 3, Thank you for your comment and we have revised the submission to better reflect the need for further Urecholine research.

I agree with your suggestion that Urecholine can be consider as a useful agent for improving decatherization.
